# SimplE Embedding for Link Prediction in Knowledge Graphs

**Seyed Mehran Kazemi**
University of British Columbia
Vancouver, BC, Canada
smkazemi@cs.ubc.ca

**David Poole**
University of British Columbia
Vancouver, BC, Canada
poole@cs.ubc.ca

## Abstract

Knowledge graphs contain knowledge about the world and provide a structured representation of this knowledge. Current knowledge graphs contain only a small subset of what is true in the world. *Link prediction* approaches aim at predicting new links for a knowledge graph given the existing links among the entities. *Tensor factorization* approaches have proved promising for such link prediction problems. Proposed in 1927, Canonical Polyadic (CP) decomposition is among the first tensor factorization approaches. CP generally performs poorly for link prediction as it learns two independent embedding vectors for each entity, whereas they are really tied. We present a simple enhancement of CP (which we call *SimplE*) to allow the two embeddings of each entity to be learned dependently. The complexity of *SimplE* grows linearly with the size of embeddings. The embeddings learned through *SimplE* are interpretable, and certain types of background knowledge can be incorporated into these embeddings through weight tying. We prove *SimplE* is fully expressive and derive a bound on the size of its embeddings for full expressivity. We show empirically that, despite its simplicity, *SimplE* outperforms several state-of-the-art tensor factorization techniques. SimplE's code is available on GitHub at https://github.com/Mehran-k/SimplE.

## 1 Introduction

During the past two decades, several knowledge graphs (KGs) containing (perhaps probabilistic) facts about the world have been constructed. These KGs have applications in several fields including search, question answering, natural language processing, recommendation systems, etc. Due to the enormous number of facts that could be asserted about our world and the difficulty in accessing and storing all these facts, KGs are incomplete. However, it is possible to predict new links in a KG based on the existing ones. *Link prediction* and several other related problems aiming at reasoning with entities and relationships are studied under the umbrella of *statistical relational learning (SRL)* [12, 31, 7]. The problem of link prediction for KGs is also known as *knowledge graph completion*. A KG can be represented as a set of $(head, relation, tail)$ triples[1]. The problem of KG completion can be viewed as predicting new triples based on the existing ones.

Tensor factorization approaches have proved to be an effective SRL approach for KG completion [29, 4, 39, 26]. These approaches consider embeddings for each entity and each relation. To predict whether a triple holds, they use a function which takes the embeddings for the head and tail entities and the relation as input and outputs a number indicating the predicted probability. Details and discussions of these approaches can be found in several recent surveys [27, 43].

One of the first tensor factorization approaches is the *canonical Polyadic (CP)* decomposition [15]. This approach learns one embedding vector for each relation and two embedding vectors for each entity, one to be used when the entity is the *head* and one to be used when the entity is the *tail*. The head embedding of an entity is learned independently of (and is unrelated to) its tail embedding. This independence has caused CP to perform poorly for KG completion [40]. In this paper, we develop a tensor factorization approach based on CP that addresses the independence among the two embedding vectors of the entities. Due to the simplicity of our model, we call it *SimplE* (**Simpl**e **E**mbedding).

We show that *SimplE*: 1- can be considered a bilinear model, 2- is fully expressive, 3- is capable of encoding background knowledge into its embeddings through parameter sharing (aka weight tying), and 4- performs very well empirically despite (or maybe because of) its simplicity. We also discuss several disadvantages of other existing approaches. We prove that many existing translational approaches (see e.g., [4, 17, 41, 26]) are not fully expressive and we identify severe restrictions on what they can represent. We also show that the function used in *ComplEx* [39, 40], a state-of-the-art approach for link prediction, involves redundant computations.

## 2 Background and Notation

We represent vectors with lowercase letters and matrices with uppercase letters. Let $v, w, x \in \mathbb{R}^d$ be vectors of length $d$. We define $\langle v, w, x \rangle \doteq \sum_{j=1}^{d} v[j] * w[j] * x[j]$, where $v[j]$, $w[j]$, and $x[j]$ represent the $j$th element of $v$, $w$ and $x$ respectively. That is, $\langle v, w, x \rangle \doteq (v \odot w) \cdot x$ where $\odot$ represents element-wise (Hadamard) multiplication and $\cdot$ represents dot product. $\mathcal{I}^d$ represents an identity matrix of size $d$. $[v_1; v_2; \ldots; v_n]$ represents the concatenation of $n$ vectors $v_1$, $v_2$, $\ldots$ and $v_n$.

Let $\mathcal{E}$ and $\mathcal{R}$ represent the set of entities and relations respectively. A **triple** is represented as $(h, r, t)$, where $h \in \mathcal{E}$ is the *head*, $r \in \mathcal{R}$ is the relation, and $t \in \mathcal{E}$ is the *tail* of the triple. Let $\zeta$ represent the set of all triples that are true in a world (e.g., $(paris, capitalOf, france)$), and $\zeta'$ represent the ones that are false (e.g., $(paris, capitalOf, italy)$). A **knowledge graph** $\mathcal{KG}$ is a subset of $\zeta$. A relation $r$ is **reflexive** on a set $\mathcal{E}$ of entities if $(e, r, e) \in \zeta$ for all entities $e \in \mathcal{E}$. A relation $r$ is **symmetric** on a set $\mathcal{E}$ of entities if $(e_1, r, e_2) \in \zeta \iff (e_2, r, e_1) \in \zeta$ for all pairs of entities $e_1, e_2 \in \mathcal{E}$, and is **anti-symmetric** if $(e_1, r, e_2) \in \zeta \iff (e_2, r, e_1) \in \zeta'$. A relation $r$ is **transitive** on a set $\mathcal{E}$ of entities if $(e_1, r, e_2) \in \zeta \land (e_2, r, e_3) \in \zeta \Rightarrow (e_1, r, e_3) \in \zeta$ for all $e_1, e_2, e_3 \in \mathcal{E}$. The inverse of a relation $r$, denoted as $r^{-1}$, is a relation such that for any two entities $e_i$ and $e_j$, $(e_i, r, e_j) \in \zeta \iff (e_j, r^{-1}, e_i) \in \zeta$.

An **embedding** is a function from an entity or a relation to one or more vectors or matrices of numbers. A **tensor factorization** model defines two things: 1- the embedding functions for entities and relations, 2- a function $f$ taking the embeddings for $h$, $r$ and $t$ as input and generating a prediction of whether $(h, r, t)$ is in $\zeta$ or not. The values of the embeddings are learned using the triples in a $\mathcal{KG}$. A tensor factorization model is **fully expressive** if given any ground truth (full assignment of truth values to all triples), there exists an assignment of values to the embeddings of the entities and relations that accurately separates the correct triples from incorrect ones.

## 3 Related Work

**Translational Approaches** define additive functions over embeddings. In many translational approaches, the embedding for each entity $e$ is a single vector $v_e \in \mathbb{R}^d$ and the embedding for each relation $r$ is a vector $v_r \in \mathbb{R}^{d'}$ and two matrices $P_r \in \mathbb{R}^{d' \times d}$ and $Q_r \in \mathbb{R}^{d' \times d}$. The dissimilarity function for a triple $(h, r, t)$ is defined as $||P_r v_h + v_r - Q_r v_t||_i$ (i.e. encouraging $P_r v_h + v_r \approx Q_r v_t$) where $||v||_i$ represents norm $i$ of vector $v$. Translational approaches having this dissimilarity function usually differ on the restrictions they impose on $P_r$ and $Q_r$. In TransE [4], $d = d'$, $P_r = Q_r = \mathcal{I}^d$. In TransR [22], $P_r = Q_r$. In STransE [26], no restrictions are imposed on the matrices. FTransE [11], slightly changes the dissimilarity function defining it as $||P_r v_h + v_r - \alpha Q_r v_t||_i$ for a value of $\alpha$ that minimizes the norm for each triple. In the rest of the paper, we let *FSTransE* represent the FTransE model where no restrictions are imposed over $P_r$ and $Q_r$.

**Multiplicative Approaches** define product-based functions over embeddings. DistMult [46], one of the simplest multiplicative approaches, considers the embeddings for each entity and each relation to be $v_e \in \mathbb{R}^d$ and $v_r \in \mathbb{R}^d$ respectively and defines its similarity function for a triple $(h, r, t)$

as $\langle v_h, v_r, v_t \rangle$. Since DistMult does not distinguish between head and tail entities, it can only model symmetric relations. ComplEx [39] extends DistMult by considering complex-valued instead of real-valued vectors for entities and relations. For each entity $e$, let $re_e \in \mathbb{R}^d$ and $im_e \in \mathbb{R}^d$ represent the real and imaginary parts of the embedding for $e$. For each relation $r$, let $re_r \in \mathbb{R}^d$ and $im_r \in \mathbb{R}^d$ represent the real and imaginary parts of the embedding for $r$. Then the similarity function of ComplEx for a triple $(h, r, t)$ is defined as $Real(\sum_{j=1}^{d}(re_h[j] + im_h[j]i) * (re_r[j] + im_r[j]i) * (re_t[j] - im_t[j]i))$, where $Real(\alpha + \beta i) = \alpha$ and $i^2 = -1$. One can easily verify that the function used by ComplEx can be expanded and written as $\langle re_h, re_r, re_t \rangle + \langle re_h, im_r, im_t \rangle + \langle im_h, re_r, im_t \rangle - \langle im_h, im_r, re_t \rangle$. In RESCAL [28], the embedding vector for each entity $e$ is $v_e \in \mathbb{R}^d$ and for each relation $r$ is $v_r \in \mathbb{R}^{d \times d}$ and the similarity function for a triple $(h, r, t)$ is $v_r \cdot vec(v_h \otimes v_t)$, where $\otimes$ represents the outer product of two vectors and $vec(.)$ vectorizes the input matrix. HolE [32] is a multiplicative model that is isomorphic to ComplEx [14].

**Deep Learning Approaches** generally use a neural network that learns how the head, relation, and tail embeddings interact. E-MLP [37] considers the embeddings for each entity $e$ to be a vector $v_e \in \mathbb{R}^d$, and for each relation $r$ to be a matrix $M_r \in \mathbb{R}^{2k \times m}$ and a vector $v_r \in \mathbb{R}^m$. To make a prediction about a triple $(h, r, t)$, E-MLP feeds $[v_h; v_t] \in \mathbb{R}^{2d}$ into a two-layer neural network whose weights for the first layer are the matrix $M_r$ and for the second layer are $v_r$. ER-MLP [10], considers the embeddings for both entities and relations to be single vectors and feeds $[v_h; v_r; v_t] \in \mathbb{R}^{3d}$ into a two layer neural network. In [35], once the entity vectors are provided by the convolutional neural network and the relation vector is provided by the long-short time memory network, for each triple the vectors are concatenated similar to ER-MLP and are fed into a four-layer neural network. Neural tensor network (NTN) [37] combines E-MLP with several bilinear parts (see Subsection 5.4 for a definition of bilinear models).

# 4    SimplE: A Simple Yet Fully Expressive Model

In *canonical Polyadic (CP)* decomposition [15], the embedding for each entity $e$ has two vectors $h_e, t_e \in \mathbb{R}^d$, and for each relation $r$ has a single vector $v_r \in \mathbb{R}^d$. $h_e$ captures $e$'s behaviour as the head of a relation and $t_e$ captures $e$'s behaviour as the tail of a relation. The similarity function for a triple $(e_1, r, e_2)$ is $\langle h_{e_1}, v_r, t_{e_2} \rangle$. In CP, the two embedding vectors for entities are learned independently of each other: observing $(e_1, r, e_2) \in \zeta$ only updates $h_{e_1}$ and $t_{e_2}$, not $t_{e_1}$ and $h_{e_2}$.

**Example 1.** Let $likes(p, m)$ represent if a person $p$ likes a movie $m$ and $acted(m, a)$ represent who acted in which movie. Which actors play in a movie is expected to affect who likes the movie. In CP, observations about *likes* only update the $t$ vector of movies and observations about *acted* only update the $h$ vector. Therefore, what is being learned about movies through observations about *acted* does not affect the predictions about *likes* and vice versa.

*SimplE* takes advantage of the inverse of relations to address the independence of the two vectors for each entity in CP. While inverse of relations has been used for other purposes (see e.g., [20, 21, 6]), using them to address the independence of the entity vectors in CP is a novel contribution.

**Model Definition:** SimplE considers two vectors $h_e, t_e \in \mathbb{R}^d$ as the embedding of each entity $e$ (similar to CP), and two vectors $v_r, v_{r^{-1}} \in \mathbb{R}^d$ for each relation $r$. The similarity function of SimplE for a triple $(e_i, r, e_j)$ is defined as $\frac{1}{2}(\langle h_{e_i}, v_r, t_{e_j} \rangle + \langle h_{e_j}, v_{r^{-1}}, t_{e_i} \rangle)$, i.e. the average of the CP scores for $(e_i, r, e_j)$ and $(e_j, r^{-1}, e_i)$. In our experiments, we also consider a different variant, which we call *SimplE-ignr*. During training, for each correct (incorrect) triple $(e_i, r, e_j)$, SimplE-ignr updates the embeddings such that each of the two scores $\langle h_{e_i}, v_r, t_{e_j} \rangle$ and $\langle h_{e_j}, v_{r^{-1}}, t_{e_i} \rangle$ become larger (smaller). During testing, *SimplE-ignr* ignores $r^{-1}s$ and defines the similarity function to be $\langle h_{e_i}, v_r, t_{e_j} \rangle$.

**Learning SimplE Models:** To learn a SimplE model, we use stochastic gradient descent with mini-batches. In each learning iteration, we iteratively take in a batch of positive triples from the $\mathcal{KG}$, then for each positive triple in the batch we generate $n$ negative triples by corrupting the positive triple. We use Bordes *et al.* [4]'s procedure to corrupt positive triples. The procedure is as follows. For a positive triple $(h, r, t)$, we randomly decide to corrupt the head or tail. If the head is selected, we replace $h$ in the triple with an entity $h'$ randomly selected from $\mathcal{E} - \{h\}$ and generate the corrupted triple $(h', r, t)$. If the tail is selected, we replace $t$ in the triple with an entity $t'$ randomly selected from $\mathcal{E} - \{t\}$ and generate the corrupted triple $(h, r, t')$. We generate a labelled batch **LB** by labelling positive triples as

Figure 1: $h_e$s and $v_r$s in the proof of Proposition 1.

| | | | | | | | | | | | | | | | | |
|---|---|---|---|---|---|---|---|---|---|---|---|---|---|---|---|---|
| $h(e_0)$ | 1 | 0 | 0 | ... | 0 | 1 | 0 | 0 | ... | 0 | ... | 1 | 0 | 0 | ... | 0 |
| $h(e_1)$ | 0 | 1 | 0 | ... | 0 | 0 | 1 | 0 | ... | 0 | ... | 0 | 1 | 0 | ... | 0 |
| $h(e_2)$ | 0 | 0 | 1 | ... | 0 | 0 | 0 | 1 | ... | 0 | ... | 0 | 0 | 1 | ... | 0 |
| ... | | | | | | | | | | | | | | | | |
| $h(e_{|\mathcal{E}|-1})$ | 0 | 0 | 0 | ... | 1 | 0 | 0 | 0 | ... | 1 | ... | 0 | 0 | 0 | ... | 1 |
| $v(r_0)$ | 1 | 1 | 1 | ... | 1 | 0 | 0 | 0 | ... | 0 | ... | 0 | 0 | 0 | ... | 0 |
| $v(r_1)$ | 0 | 0 | 0 | ... | 0 | 1 | 1 | 1 | ... | 1 | ... | 0 | 0 | 0 | ... | 0 |
| ... | | | | | | | | | | | | | | | | |
| $v(r_{|\mathcal{R}|-1})$ | 0 | 0 | 0 | ... | 0 | 0 | 0 | 0 | ... | 0 | ... | 1 | 1 | 1 | ... | 1 |

$+1$ and negatives as $-1$. Once we have a labelled batch, following [39] we optimize the $L2$ regularized negative log-likelihood of the batch: $\min_\theta \sum_{((h,r,t),l)\in \mathbf{LB}} softplus(-l \cdot \phi(h,r,t)) + \lambda||\theta||_2^2$, where $\theta$ represents the parameters of the model (the parameters in the embeddings), $l$ represents the label of a triple, $\phi(h,r,t)$ represents the similarity score for triple $(h,r,t)$, $\lambda$ is the regularization hyper-parameter, and $softplus(x) = log(1+\exp(x))$. While several previous works (e.g., TransE, TransR, STransE, etc.) consider a margin-based loss function, Trouillon and Nickel [38] show that the margin-based loss function is more prone to overfitting compared to log-likelihood.

# 5 Theoretical Analyses

In this section, we provide some theoretical analyses of SimplE and other existing approaches.

## 5.1 Fully Expressiveness

The following proposition establishes the full expressivity of SimplE.

**Proposition 1.** *For any ground truth over entities $\mathcal{E}$ and relations $\mathcal{R}$ containing $\gamma$ true facts, there exists a SimplE model with embedding vectors of size $min(|\mathcal{E}| \cdot |\mathcal{R}|, \gamma + 1)$ that represents that ground truth.*

*Proof.* First, we prove the $|\mathcal{E}| \cdot |\mathcal{R}|$ bound. With embedding vectors of size $|\mathcal{E}| * |\mathcal{R}|$, for each entity $e_i$ we let the n-th element of $h_{e_i} = 1$ if $(n \bmod |\mathcal{E}|) = i$ and 0 otherwise, and for each relation $r_j$ we let the n-th element of $v_{r_j} = 1$ if $(n \ div \ |\mathcal{E}|) = j$ and 0 otherwise (see Fig 1). Then for each $e_i$ and $r_j$, the product of $h_{e_i}$ and $v_{r_j}$ is 0 everywhere except for the $(j * |\mathcal{E}| + i)$-th element. So for each entity $e_k$, we set the $(j * |\mathcal{E}| + i)$-th element of $t_{e_k}$ to be 1 if $(e_i, r_j, e_k)$ holds and $-1$ otherwise.

Now we prove the $\gamma + 1$ bound. Let $\gamma$ be zero (base of the induction). We can have embedding vectors of size 1 for each entity and relation, setting the value for entities to 1 and for relations to $-1$. Then $\langle h_{e_i}, v_{r_j}, t_{e_k} \rangle$ is negative for every entities $e_i$ and $e_k$ and relation $r_j$. So there exists embedding vectors of size $\gamma + 1$ that represents this ground truth. Let us assume for any ground truth where $\gamma = n - 1$ $(1 \le n \le |\mathcal{R}||\mathcal{E}|^2)$, there exists an assignment of values to embedding vectors of size $n$ that represents that ground truth (assumption of the induction). We must prove for any ground truth where $\gamma = n$, there exists an assignment of values to embedding vectors of size $n + 1$ that represents this ground truth. Let $(e_i, r_j, e_k)$ be one of the $n$ true facts. Consider a modified ground truth which is identical to the ground truth with $n$ true facts, except that $(e_i, r_j, e_k)$ is assigned false. The modified ground truth has $n - 1$ true facts and based on the assumption of the induction, we can represent it using some embedding vectors of size $n$. Let $q = \langle h_{e_i}, v_{r_j}, t_{e_k} \rangle$ where $h_{e_i}, v_{r_j}$ and $t_{e_k}$ are the embedding vectors that represent the modified ground truth. We add an element to the end of all embedding vectors and set it to 0. This increases the vector sizes to $n + 1$ but does not change any scores. Then we set the last element of $h_{e_i}$ to 1, $v_{r_j}$ to 1, and $t_{e_k}$ to $q + 1$. This ensures that $\langle h_{e_i}, v_{r_j}, t_{e_k} \rangle > 0$ for the new vectors, and no other score is affected. □

DistMult is not fully expressive as it forces relations to be symmetric. It has been shown in [40] that ComplEx is fully expressive with embeddings of length at most $|\mathcal{E}| \cdot |\mathcal{R}|$. According to the universal approximation theorem [5, 16], under certain conditions, neural networks are universal approximators of continuous functions over compact sets. Therefore, we would expect there to be a representation

based on neural networks that can approximate any ground truth, but the number of hidden units might have to grow with the number of triples. Wang *et al.* [44] prove that *TransE* is not fully expressive. Proposition 2 proves that not only TransE but also many other translational approaches are not fully expressive. The proposition also identifies severe restrictions on what relations these approaches can represent.

**Proposition 2.** *FSTransE is not fully expressive and has the following restrictions.* R1 : *If a relation $r$ is reflexive on $\Delta \subset \mathcal{E}$, $r$ must also be symmetric on $\Delta$,* R2 : *If $r$ is reflexive on $\Delta \subset \mathcal{E}$, $r$ must also be transitive on $\Delta$, and* R3 : *If entity $e_1$ has relation $r$ with every entity in $\Delta \subset \mathcal{E}$ and entity $e_2$ has relation $r$ with one of the entities in $\Delta$, then $e_2$ must have the relation $r$ with every entity in $\Delta$.*

*Proof.* For any entity $e$ and relation $r$, let $p_{re} = P_r v_e$ and $q_{re} = Q_r v_e$. For a triple $(h, r, t)$ to hold, we should ideally have $p_{rh} + v_r = \alpha q_{rt}$ for some $\alpha$. We assume $s_1$, $s_2$, $s_3$ and $s_4$ are entities in $\Delta$.

R1 : A relation $r$ being reflexive on $\Delta$ implies $p_{rs_1} + v_r = \alpha_1 q_{rs_1}$ and $p_{rs_2} + v_r = \alpha_2 q_{rs_2}$. Suppose $(s_1, r, s_2)$ holds as well. Then we know $p_{rs_1} + v_r = \alpha_3 q_{rs_2}$. Therefore, $p_{rs_2} + v_r = \alpha_2 q_{rs_2} = \frac{\alpha_2}{\alpha_3}(p_{rs_1} + v_r) = \frac{\alpha_2}{\alpha_3}\alpha_1 q_{rs_1} = \alpha_4 q_{rs_1}$, where $\alpha_4 = \frac{\alpha_2 \alpha_1}{\alpha_3}$. Therefore, $(s_2, r, s_1)$ must holds.

R2 : A relation $r$ being reflexive implies $p_{rs_1} + v_r = \alpha_1 q_{rs_1}$, $p_{rs_2} + v_r = \alpha_2 q_{rs_2}$, and $p_{rs_3} + v_r = \alpha_3 q_{rs_3}$. Suppose $(s_1, r, s_2)$ and $(s_2, r, s_3)$ hold. Then we know $p_{rs_1} + v_r = \alpha_4 q_{rs_2}$ and $p_{rs_2} + v_r = \alpha_5 q_{rs_3}$. We can conclude $p_{rs_1} + v_r = \alpha_4 q_{rs_2} = \frac{\alpha_4}{\alpha_2}(p_{rs_2} + v_r) = \frac{\alpha_4}{\alpha_2}\alpha_5 q_{rs_3} = \alpha_6 q_{rs_3}$, where $\alpha_6 = \frac{\alpha_4 \alpha_5}{\alpha_2}$. The above equality proves $(s_1, r, s_3)$ must hold.

R3 : Let $e_2$ have relation $r$ with $s_1$. We know $p_{re_1} + v_r = \alpha_1 q_{rs_1}$, $p_{re_1} + v_r = \alpha_2 q_{rs_2}$, and $p_{re_2} + v_r = \alpha_3 q_{rs_1}$. We can conclude $p_{re_2} + v_r = \alpha_3 q_{rs_1} = \frac{\alpha_3}{\alpha_1}(p_{re_1} + v_r) = \frac{\alpha_3}{\alpha_1}\alpha_2 q_{rs_2} = \alpha_4 q_{rs_2}$, where $\alpha_4 = \frac{\alpha_3 \alpha_2}{\alpha_1}$. Therefore, $(e_2, r, s_2)$ must hold. $\square$

**Corollary 1.** *Other variants of translational approaches such as TransE, FTransE, STransE, TransH [41], and TransR [22] also have the restrictions mentioned in Proposition 2.*

## 5.2 Incorporating Background Knowledge into the Embeddings

In SimplE, each element of the embedding vector of the entities can be considered as a feature of the entity and the corresponding element of a relation can be considered as a measure of how important that feature is to the relation. Such interpretability allows the embeddings learned through SimplE for an entity (or relation) to be potentially transferred to other domains. It also allows for incorporating observed features of entities into the embeddings by fixing one of the elements of the embedding vector of the observed value. Nickel *et al.* [30] show that incorporating such features helps reduce the size of the embeddings.

Recently, incorporating background knowledge into tensor factorization approaches has been the focus of several studies. Towards this goal, many existing approaches rely on post-processing steps or add additional terms to the loss function to penalize predictions that violate the background knowledge [34, 42, 45, 13, 9]. Minervini *et al.* [25] show how background knowledge in terms of equivalence and inversion can be incorporated into several tensor factorization models through parameter tying[2]. Incorporating background knowledge by parameter tying has the advantage of guaranteeing the predictions follow the background knowledge for all embeddings. In this section, we show how three types of background knowledge, namely symmetry, anti-symmetry, and inversion, can be incorporated into the embeddings of SimplE by tying the parameters[3] (we ignore the equivalence between two relations as it is trivial).

**Proposition 3.** *Let $r$ be a relation such that for any two entities $e_i$ and $e_j$ we have $(e_i, r, e_j) \in \zeta \iff (e_j, r, e_i) \in \zeta$ (i.e. $r$ is symmetric). This property of $r$ can be encoded into SimplE by tying the parameters $v_{r^{-1}}$ to $v_r$.*

*Proof.* If $(e_i, r, e_j) \in \zeta$, then a SimplE model makes $\langle h_{e_i}, v_r, t_{e_j} \rangle$ and $\langle h_{e_j}, v_{r^{-1}}, t_{e_i} \rangle$ positive. By tying the parameters $v_{r^{-1}}$ to $v_r$, we can conclude that $\langle h_{e_j}, v_r, t_{e_i} \rangle$ and $\langle h_{e_i}, v_{r^{-1}}, t_{e_j} \rangle$ also become positive. Therefore, the SimplE model predicts $(e_j, r, e_i) \in \zeta$. $\square$

**Proposition 4.** *Let $r$ be a relation such that for any two entities $e_i$ and $e_j$ we have $(e_i, r, e_j) \in \zeta \iff (e_j, r, e_i) \in \zeta'$ (i.e. $r$ is anti-symmetric). This property of $r$ can be encoded into SimplE by tying the parameters $v_{r^{-1}}$ to the negative of $v_r$.*

*Proof.* If $(e_i, r, e_j) \in \zeta$, then a SimplE model makes $\langle h_{e_i}, v_r, t_{e_j} \rangle$ and $\langle h_{e_j}, v_{r^{-1}}, t_{e_i} \rangle$ positive. By tying the parameters $v_{r^{-1}}$ to the negative of $v_r$, we can conclude that $\langle h_{e_j}, v_r, t_{e_i} \rangle$ and $\langle h_{e_i}, v_{r^{-1}}, t_{e_j} \rangle$ become negative. Therefore, the SimplE model predicts $(e_j, r, e_i) \in \zeta'$. $\qquad\square$

**Proposition 5.** *Let $r_1$ and $r_2$ be two relations such that for any two entities $e_i$ and $e_j$ we have $(e_i, r_1, e_j) \in \zeta \iff (e_j, r_2, e_i) \in \zeta$ (i.e. $r_2$ is the inverse of $r_1$). This property of $r_1$ and $r_2$ can be encoded into SimplE by tying the parameters $v_{r_1^{-1}}$ to $v_{r_2}$ and $v_{r_2^{-1}}$ to $v_{r_1}$.*

*Proof.* If $(e_i, r_1, e_j) \in \zeta$, then a SimplE model makes $\langle h_{e_i}, v_{r_1}, t_{e_j} \rangle$ and $\langle h_{e_j}, v_{r_1^{-1}}, t_{e_i} \rangle$ positive. By tying the parameters $v_{r_2^{-1}}$ to $v_{r_1}$ and $v_{r_2}$ to $v_{r_1^{-1}}$, we can conclude that $\langle h_{e_i}, v_{r_2^{-1}}, t_{e_j} \rangle$ and $\langle h_{e_j}, v_{r_2}, t_{e_i} \rangle$ also become positive. Therefore, the SimplE model predicts $(e_j, r_2, e_i) \in \zeta$. $\qquad\square$

## 5.3 Time Complexity and Parameter Growth

As described in [3], to scale to the size of the current KGs and keep up with their growth, a relational model must have a linear time and memory complexity. Furthermore, one of the important challenges in designing tensor factorization models is the trade-off between expressivity and model complexity. Models with many parameters usually overfit and give poor performance. While the time complexity for TransE is $O(d)$ where $d$ is the size of the embedding vectors, adding the projections as in STransE (through the two relation matrices) increases the time complexity to $O(d^2)$. Besides time complexity, the number of parameters to be learned from data grows quadratically with $d$. A quadratic time complexity and parameter growth may arise two issues: 1- scalability problems, 2- overfitting. Same issues exist for models such as RESCAL and NTNs that have quadratic or higher time complexities and parameter growths. DistMult and ComplEx have linear time complexities and the number of their parameters grow linearly with $d$.

The time complexity of both *SimplE-ignr* and *SimplE* is $O(d)$, i.e. linear in the size of vector embeddings. *SimplE-ignr* requires one multiplication between three vectors for each triple. This number is 2 for *SimplE* and 4 for *ComplEx*. Thus, with the same number of parameters, *SimplE-ignr* and *SimplE* reduce the computations by a factor of 4 and 2 respectively compared to *ComplEx*.

## 5.4 Family of Bilinear Models

Bilinear models correspond to the family of models where the embedding for each entity $e$ is $v_e \in \mathbb{R}^d$, for each relation $r$ is $M_r \in \mathbb{R}^{d \times d}$ (with certain restrictions), and the similarity function for a triple $(h, r, t)$ is defined as $v_h^T M_r v_t$. These models have shown remarkable performance for link prediction in knowledge graphs [31]. DistMult, ComplEx, and RESCAL are known to belong to the family of bilinear models. We show that SimplE (and CP) also belong to this family.

DistMult can be considered a bilinear model which restricts the $M_r$ matrices to be diagonal as in Fig. 2(a). For ComplEx, if we consider the embedding for each entity $e$ to be a single vector $[re_e; im_e] \in \mathbb{R}^{2d}$, then it can be considered a bilinear model with its $M_r$ matrices constrained according to Fig. 2(b). RESCAL can be considered a bilinear model which imposes no constraints on the $M_r$ matrices. Considering the embedding for each entity $e$ to be a single vector $[h_e; t_e] \in \mathbb{R}^{2d}$, CP can be viewed as a bilinear model with its $M_r$ matrices constrained as in Fig 2(c). For a triple $(e_1, r, e_2)$, multiplying $[h_{e_1}; t_{e_1}]$ to $M_r$ results in a vector $v_{e_1 r}$ whose first half is zero and whose second half corresponds to an element-wise product of $h_{e_1}$ to the parameters in $M_r$. Multiplying $v_{e_1 r}$ to $[h_{e_2}; t_{e_2}]$ corresponds to ignoring $h_{e_2}$ (since the first half of $v_{e_1 r}$ is zeros) and taking the dot-product of the second half of $v_{e_1 r}$ with $t_{e_2}$. SimplE can be viewed as a bilinear model similar to CP except that the $M_r$ matrices are constrained as in Fig 2(d). The extra parameters added to the matrix compared to CP correspond to the parameters in the inverse of the relations.

The constraint over $M_r$ matrices in SimplE is very similar to the constraint in DistMult. $v_h^T M_r$ in both SimplE and DistMult can be considered as an element-wise product of the parameters, except that the $M_r$s in SimplE swap the first and second halves of the resulting vector. Compared to ComplEx, SimplE removes the parameters on the main diagonal of $M_r$s. Note that several other restrictions on

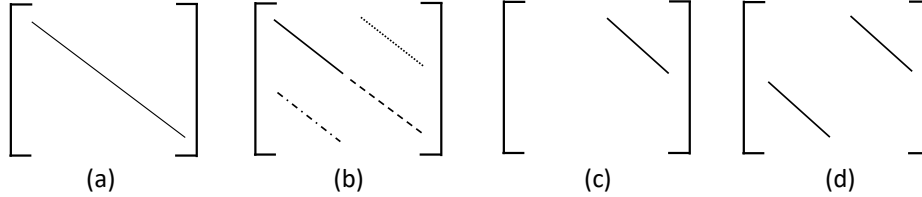

Figure 2: The constraints over $M_r$ matrices for bilinear models (a) DistMult, (b) ComplEx, (c) CP, and (d) SimplE. The lines represent where the parameters are; other elements of the matrices are constrained to be zero. In ComplEx, the parameters represented by the dashed line is tied to the parameters represented by the solid line and the parameters represented by the dotted line is tied to the negative of the dotted-and-dashed line.

the $M_r$ matrices are equivalent to SimplE. Viewing SimplE as a single-vector-per-entity model makes it easily integrable (or compatible) with other embedding models (in knowledge graph completion, computer vision and natural language processing) such as [35, 47, 36].

## 5.5  Redundancy in ComplEx

As argued earlier, with the same number of parameters, the number of computations in ComplEx are 4x and 2x more than SimplE-ignr and SimplE. Here we show that a portion of the computations performed by ComplEx to make predictions is redundant. Consider a ComplEx model with embedding vectors of size 1 (for ease of exposition). Suppose the embedding vectors for $h$, $r$ and $t$ are $[\alpha_1 + \beta_1 i]$, $[\alpha_2 + \beta_2 i]$, and $[\alpha_3 + \beta_3 i]$ respectively. Then the probability of $(h, r, t)$ being correct according to ComplEx is proportional to the sum of the following four terms: 1) $\alpha_1\alpha_2\alpha_3$, 2) $\alpha_1\beta_2\beta_3$, 3) $\beta_1\alpha_2\beta_3$, and 4) $-\beta_1\beta_2\alpha_3$. It can be verified that for any assignment of (non-zero) values to $\alpha_i$s and $\beta_i$s, at least one of the above terms is negative. This means for a correct triple, ComplEx uses three terms to overestimate its score and then uses a term to cancel the overestimation.

The following example shows how this redundancy in ComplEx may affect its interpretability:

**Example 2.** Consider a ComplEx model with embeddings of size 1. Consider entities $e_1$, $e_2$ and $e_3$ with embedding vectors $[1 + 4i]$, $[1 + 6i]$, and $[3 + 2i]$ respectively, and a relation $r$ with embedding vector $[1 + i]$. According to ComplEx, the score for triple $(e_1, r, e_3)$ is positive suggesting $e_1$ probably has relation $r$ with $e_3$. However the score for triple $(e_2, r, e_3)$ is negative suggesting $e_2$ probably does not have relation $r$ with $e_3$. Since the only difference between $e_1$ and $e_2$ is that the imaginary part changes from 4 to 6, it is difficult to associate a meaning to these numbers.

# 6  Experiments and Results

**Datasets:** We conducted experiments on two standard benchmarks: WN18 a subset of *Wordnet* [24], and FB15k a subset of *Freebase* [2]. We used the same train/valid/test sets as in [4]. WN18 contains $40,943$ entities, $18$ relations, $141,442$ train, $5,000$ validation and $5,000$ test triples. FB15k contains $14,951$ entities, $1,345$ relations, $483,142$ train, $50,000$ validation, and $59,071$ test triples.

**Baselines:** We compare SimplE with several existing tensor factorization approaches. Our baselines include canonical Polyadic (CP) decomposition, TransE, TransR, DistMult, NTN, STransE, ER-MLP, and ComplEx. Given that we use the same data splits and objective function as ComplEx, we report the results of CP, TransE, DistMult, and ComplEx from [39]. We report the results of TransR and NTN from [27], and ER-MLP from [32] for further comparison.

**Evaluation Metrics:** To measure and compare the performances of different models, for each test triple $(h, r, t)$ we compute the score of $(h', r, t)$ triples for all $h' \in \mathcal{E}$ and calculate the ranking $rank_h$ of the triple having $h$, and we compute the score of $(h, r, t')$ triples for all $t' \in \mathcal{E}$ and calculate the ranking $rank_t$ of the triple having $t$. Then we compute the *mean reciprocal rank (MRR)* of these rankings as the mean of the inverse of the rankings: $MRR = \frac{1}{2*|tt|} \sum_{(h,r,t) \in tt} \frac{1}{rank_h} + \frac{1}{rank_t}$, where $tt$ represents the test triples. MRR is a more robust measure than *mean rank*, since a single bad ranking can largely influence *mean rank*.

Table 1: Results on WN18 and FB15k. Best results are in bold.

| | WN18 | | | | | FB15k | | | | |
|---|---|---|---|---|---|---|---|---|---|---|
| | MRR | | Hit@ | | | MRR | | Hit@ | | |
| Model | Filter | Raw | 1 | 3 | 10 | Filter | Raw | 1 | 3 | 10 |
| CP | 0.075 | 0.058 | 0.049 | 0.080 | 0.125 | 0.326 | 0.152 | 0.219 | 0.376 | 0.532 |
| TransE | 0.454 | 0.335 | 0.089 | 0.823 | 0.934 | 0.380 | 0.221 | 0.231 | 0.472 | 0.641 |
| TransR | 0.605 | 0.427 | 0.335 | 0.876 | 0.940 | 0.346 | 0.198 | 0.218 | 0.404 | 0.582 |
| DistMult | 0.822 | 0.532 | 0.728 | 0.914 | 0.936 | 0.654 | 0.242 | 0.546 | 0.733 | 0.824 |
| NTN | 0.530 | – | – | – | 0.661 | 0.250 | – | – | – | 0.414 |
| STransE | 0.657 | 0.469 | – | – | 0.934 | 0.543 | **0.252** | – | – | 0.797 |
| ER-MLP | 0.712 | 0.528 | 0.626 | 0.775 | 0.863 | 0.288 | 0.155 | 0.173 | 0.317 | 0.501 |
| ComplEx | 0.941 | 0.587 | 0.936 | **0.945** | **0.947** | 0.692 | 0.242 | 0.599 | 0.759 | **0.840** |
| SimplE-ignr | 0.939 | 0.576 | 0.938 | 0.940 | 0.941 | 0.700 | 0.237 | 0.625 | 0.754 | 0.821 |
| SimplE | **0.942** | **0.588** | **0.939** | 0.944 | **0.947** | **0.727** | 0.239 | **0.660** | **0.773** | 0.838 |

Bordes *et al.* [4] identified an issue with the above procedure for calculating the MRR (hereafter referred to as *raw MRR*). For a test triple $(h, r, t)$, since there can be several entities $h' \in \mathcal{E}$ for which $(h', r, t)$ holds, measuring the quality of a model based on its ranking for $(h, r, t)$ may be flawed. That is because two models may rank the test triple $(h, r, t)$ to be second, when the first model ranks a correct triple (e.g., from train or validation set) $(h', r, t)$ to be first and the second model ranks an incorrect triple $(h'', r, t)$ to be first. Both these models will get the same score for this test triple when the first model should get a higher score. To address this issue, [4] proposed a modification to raw MRR. For each test triple $(h, r, t)$, instead of finding the rank of this triple among triples $(h', r, t)$ for all $h' \in \mathcal{E}$ (or $(h, r, t')$ for all $t' \in \mathcal{E}$), they proposed to calculate the rank among triples $(h', r, t)$ only for $h' \in \mathcal{E}$ such that $(h', r, t) \notin train \cup valid \cup test$. Following [4], we call this measure *filtered MRR*. We also report $hit@k$ measures. The $hit@k$ for a model is computed as the percentage of test triples whose ranking (computed as described earlier) is less than or equal $k$.

**Implementation:** We implemented SimplE in TensorFlow [1]. We tuned our hyper-parameters over the validation set. We used the same search grid on embedding size and $\lambda$ as [39] to make our results directly comparable to their results. We fixed the maximum number of iterations to 1000 and the number of batches to 100. We set the learning rate for WN18 to 0.1 and for FB15k to 0.05 and used *adagrad* to update the learning rate after each batch. Following [39], we generated one negative example per positive example for WN18 and 10 negative examples per positive example in FB15k. We computed the filtered MRR of our model over the validation set every 50 iterations for WN18 and every 100 iterations for $FB15k$ and selected the iteration that resulted in the best validation filtered MRR. The best embedding size and $\lambda$ values on WN18 for SimplE-ignr were 200 and 0.001 respectively, and for SimplE were 200 and 0.03. The best embedding size and $\lambda$ values on FB15k for SimplE-ignr were 200 and 0.03 respectively, and for SimplE were 200 and 0.1.

## 6.1 Entity Prediction Results

Table 1 shows the results of our experiments. It can be viewed that both SimplE-ignr and SimplE do a good job compared to the existing baselines on both datasets. On WN18, SimplE-ignr and SimplE perform as good as ComplEx, a state-of-the-art tensor factorization model. On FB15k, SimplE outperforms the existing baselines and gives state-of-the-art results among tensor factorization approaches. SimplE (and SimplE-ignr) work especially well on this dataset in terms of filtered MRR and *hit@1*, so SimplE tends to do well at having its first prediction being correct.

The table shows that models with many parameters (e.g., NTN and STransE) do not perform well on these datasets, as they probably overfit. Translational approaches generally have an inferior performance compared to other approaches partly due to their representation restrictions mentioned in Proposition 2. As an example for the *friendship* relation in FB15k, if an entity $e_1$ is friends with 20 other entities and another entity $e_2$ is friends with only one of those 20, then according to Proposition 2 translational approaches force $e_2$ to be friends with the other 19 entities as well (same goes for, e.g., *netflix genre* in FB15k and *has part* in WN18). The table also shows that bilinear approaches tend to have better performances compared to translational and deep learning approaches. Even DistMult, the simplest bilinear approach, outperforms many translational and deep learning approaches despite not being fully expressive. We believe the simplicity of embeddings and the scoring function is a key property for the success of SimplE.

Table 2: Background Knowledge Used in Section 6.2.

| Rule Number | Rule |
|---|---|
| 1 | $(e_i, hyponym, e_j) \in \zeta \Leftrightarrow (e_j, hypernym, e_i) \in \zeta$ |
| 2 | $(e_i, memberMeronym, e_j) \in \zeta \Leftrightarrow (e_j, memberHolonym, e_i) \in \zeta$ |
| 3 | $(e_i, instanceHyponym, e_j) \in \zeta \Leftrightarrow (e_j, instanceHypernym, e_i) \in \zeta$ |
| 4 | $(e_i, hasPart, e_j) \in \zeta \Leftrightarrow (e_j, partOf, e_i) \in \zeta$ |
| 5 | $(e_i, memberOfDomainTopic, e_j) \in \zeta \Leftrightarrow (e_j, synsetDomainTopicOf, e_i) \in \zeta$ |
| 6 | $(e_i, memberOfDomainUsage, e_j) \in \zeta \Leftrightarrow (e_j, synsetDomainUsageOf, e_i) \in \zeta$ |
| 7 | $(e_i, memberOfDomainRegion, e_j) \in \zeta \Leftrightarrow (e_j, synsetDomainRegionOf, e_i) \in \zeta$ |
| 8 | $(e_i, similarTo, e_j) \in \zeta \Leftrightarrow (e_j, similarTo, e_i) \in \zeta$ |

## 6.2 Incorporating background knowledge

When background knowledge is available, we might expect that a knowledge graph might not include redundant information because it is implied by background knowledge and so the methods that do not include the background knowledge can never learn it. In section 5.2, we showed how background knowledge that can be formulated in terms of three types of rules can be incorporated into SimplE embeddings. To test this empirically, we conducted an experiment on WN18 in which we incorporated several such rules into the embeddings as outlined in Propositions 3, 4, and 5. The rules can be found in Table 2. As can be viewed in Table 2, most of the rules are of the form $\forall e_i, e_j \in \mathcal{E} : (e_i, r_1, e_j) \in \zeta \Leftrightarrow (e_j, r_2, e_i) \in \zeta$. For (possibly identical) relations such as $r_1$ and $r_2$ participating in such a rule, if both $(e_i, r_1, e_j)$ and $(e_j, r_2, e_i)$ are in the training set, one of them is redundant because one can be inferred from the other. We removed redundant triples from the training set by randomly removing one of the two triples in the training set that could be inferred from the other one based on the background rules. Removing redundant triples reduced the number of triples in the training set from (approximately) $141K$ to (approximately) $90K$, almost $36\%$ reduction in size. Note that this experiment provides an upper bound on how much background knowledge can improve the performance of a SimplE model.

We trained SimplE-ignr and SimplE (with tied parameters according to the rules) on this new training dataset with the best hyper-parameters found in the previous experiment. We refer to these two models as *SimplE-ignr-bk* and *SimplE-bk*. We also trained another SimplE-ignr and SimplE models on this dataset, but without incorporating the rules into the embeddings. For sanity check, we also trained a ComplEx model over this new dataset. We found that the filtered MRR for SimplE-ignr, SimplE, and ComplEx were respectively 0.221, 0.384, and 0.275. For SimplE-ignr-bk and SimplE-bk, the filtered MRRs were 0.772 and 0.776 respectively, substantially higher than the case without background knowledge. In terms of $hit@k$ measures, SimplE-ignr gave 0.219, 0.220, and 0.224 for $hit@1$, $hit@3$ and $hit@10$ respectively. These numbers were 0.334, 0.404, and 0.482 for SimplE, and 0.254, 0.280 and 0.313 for ComplEx. For SimplE-ignr-bk, these numbers were 0.715, 0.809 and 0.877 and for SimplE-bk they were 0.715, 0.818 and 0.883, also substantially higher than the models without background knowledge. The obtained results validate that background knowledge can be effectively incorporated into SimplE embeddings to improve its performance.

## 7  Conclusion

We proposed a simple interpretable fully expressive bilinear model for knowledge graph completion. We showed that our model, called SimplE, performs very well empirically and has several interesting properties. For instance, three types of background knowledge can be incorporated into SimplE by tying the embeddings. In future, SimplE could be improved or may help improve relational learning in several ways including: 1- building ensembles of SimplE models as [18] do it for DistMult, 2- adding SimplE to the relation-level ensembles of [44], 3- explicitly modelling the analogical structures of relations as in [23], 4- using [8]'s 1-N scoring approach to generate many negative triples for a positive triple (Trouillon *et al.* [39] show that generating more negative triples improves accuracy), 5- combining SimplE with symbolic approaches (e.g., with [19]) to improve property prediction, 6- combining SimplE with (or use SimplE as a sub-component in) techniques from other categories of relational learning as [33] do with ComplEx, 7- incorporating other types of background knowledge (e.g., entailment) into SimplE embeddings.

## Footnotes

[1]Triples are complete for relations. They are sometimes written as $(subject, verb, object)$ or $(individual, property, value)$.

[2]Although their incorporation of inversion into DistMult is not correct as it has side effects.

[3]Note that such background knowledge can be exerted on some relations selectively and not on the others. This is different than, e.g., DistMult which enforces symmetry on all relations.

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
