[Reviews · NeurIPS 2018]

Reviewer 1



The paper provides a new tensor factorization approach for knowledge graph completion. The problem is significant and an important one in AI. The main contribution is a new form of factorization that extends canonical polyadic factorization by accounting for dependence between the vectors representing the entities across relations. The proposed method seems simple to implement and also can be interpreted. Theoretical analysis is provided to establish expressivity in terms of a bound on vector size. Further, it is possible to add logical constraints to the model to represent real-world phenomena. Overall the paper is written very well and is quite easy to understand. In terms of novelty, there seems to be sufficient novelty even though it is an extension of a previous approach. The experiments are performed on two benchmarks. The only issue is the results do not show very significant improvements over state-of-the-art ComplEx. However, when background knowledge is needed the proposed approach seems better, for a modified dataset that explicitly encodes some constraints. Overall it would be nice to know the advantages of the proposed method over the state-of-the-art system. For example, are the expressiveness guarantees or ability to add logical constraints not present in Complex) . In summary though, this seems like a solid contribution for relational learning.

Reviewer 2



This paper studies the problem of knowledge graph completion. A simple approach called SimplE is proposed. SimplE is an extension of CP. The difference lies in that CP only models a given triple (h,r,t) while SimplE models both (h,r,t) and its inverse triple (t, r^{-1}, h). Strengths: 1. A simple approach with surprisingly good performance. It is surprising that just by introducing inverse triples, the link prediction performance can be improved that much (CP vs. SimplE). 2. This paper provides a thorough analysis about the newly proposed and previous emedding models (expressiveness, complexity, interpretability, and redundancy), which will help the authors to better understand these models. Weaknesses: 1. The only difference between SimplE and CP seems to be that SimplE further uses inverse triples. The technical contributions of this paper are quite limited. 2. Introducing inverse triples might also be used in other embedding models besides CP. But the authors did not test such cases in their experiments. 3. It would be better if the authors could further explain why such a simple step (introducing inverse triples) can improve the CP model that much.

Reviewer 3



The paper provides a very nice summary of the state-of-the-art and cites all the relevant publications. So, clearly the authors know the literature. They then provide a very nice and simple model that outperforms the other competing models or matches performance, while being simpler. The only additional thing I would have liked to see is an analysis of where their model outperforms other competing models. And where they might not.

Reviewer 4



This is a really nice presentation on relational learning with tensors. It has two major contributions: - a very good area survey, which nicely explains & demythifies this subject. - the simple alg, which basically adds inverse relations to improve tensor learning. The contribution itself is a bit rushed, especially on the use of rules. I don think NIPS is in the survey business, but this one does make the authorś idea very clear. Issues: Table 2 should refer the source of the results. Also: - has been there progress in any of the systems that could affect results? - the results compared to complex are really head-to-head. I was curious on how many parameters do both approaches have? - how fast do you converge? Does using inverse edges slow down convergence? - BK: itś really hard to get a hold of how far you can go using BK, that text needs cleaning up. - Is the code available?

Reviewer 5



# Summary This paper considers link prediction in knowledge graphs: give a triple (head, relation, tail) ((h,r,t)) the task is to determine whether the triple represents a correct relation. The contribution is a tensor factorization method that represents each object and relation as some vector. The main novelty relative to previous approaches is that each relation r is represented by two embedding vectors: one for r, and one for r^-1. The motivation is that should allow objects that appear as both heads and tails to more easily learn jointly from both roles. # Strengths The individual components of the paper are mostly well written and easy to understand. The core idea is simple and intuitive, and the authors do a reasonable job of motivating it. # Criticisms 1. It appears that some of the statements made in the paper are incorrect. (i) the authors claim (comparing TransE and FTransE) that a model which assigns relations based on v(t) - v(r) is less expressive than one that assigns relations based only on (v(t)-v(r))/||v(t) - v(r)||. Consider the case where the only possible embeddings are (0,0), (1,1) and (2,2). The reversal of this inclusion makes it unclear whether Corollary 1 holds---it's at least not obvious to me. (ii) At line 135, the sentence beginning "Models based on..." is not obviously true. For neural net universality to (trivially) hold all of the (h,r,t) would need to be input to some neural net---but that doesn't match the architectures described in "Deep Learning Approaches" (iii) at lines 284-285, the loss function on minibatches is given and described as a negative log likelihood. Although the expression could be considered a negative llhd for some probabilistic model, such a model has not been previously introduced. (I'm actually unsure whether the expression literally corresponds to any distribution since there's no obvious normalization term) 2. I'm not sure about the significance. In particular, does it matter that SimplE is moderately more computationally efficient than ComplEx? What does that mean for training time? For prediction time? 3. The structuring of the paper is a rather odd. My two biggest complaints here are (i) the main idea of the paper isn't given until lines 200-205 on page 5, but this could have easily been explained in the introduction (along with an overview of the full method) (ii) the loss function and training procedure are described in "Learning SimplE Models" (starting at line 275) as a subsection of "Experiments and Results". This material deals with specifying the model and should appear earlier, more prominently, and certainly not as a subsection of experiments. 4. I find the footnote on page 7 promising to release the code after the paper is submitted to be somewhat troubling. Why not include it with the submission? Although this is not strictly required, it's certainly better practice. 5. The major motivation for SimplE is that it allows the head and tail vectors for each object to be learned dependently. I don't see how this is reflected in the experiments. ### Update The authors mostly did a good job addressing my concerns (especially given the very limited space!). I still think 1.i is a problem---the fact that TransE takes embedding vectors to be points and FTransE takes embedding vectors to be lines means that TransE is more expressive. I'm also still not convinced that 'fewer multiplications' is a meaningful advantage, but the paper is nice even if this is meaningless, so whatever.

Reviewer 6



The authors present a very interesting work on link prediction in knowledge graphs using embeddings. The main idea is to provide a embedding to a reciprocal relation, which allows the model to be fully expressive while keeping it as simple as possible. The paper is very nicely written and it was a pleasure to read it (which is unfortunately becoming a rarity these days). The authors did a very good field overview, and shown benefits of the proposed approach over a number of baselines. - At first glance and without reading the paper, uppercase E in the title seems like a typo. Not sure what can be done there, but would be good to potentially reconsider the title to avoid that confusion. - Typo: Line 68, "" -> "v, w, x" - Typo: Line 151, "have and keep up", it is not clear what the authors wanted to say here, please fix. - Some definitions are repeated in the text, such as relation being symmetric. This can be cleaned up. - Line 195 paragraph, from their explanation it is not clear what the issue is. Seems reasonable not to update the same vector for "acted" and "likes", as the relations are quite different (one referring to an actor and the other to a viewer). Would be good to elaborate on this, and either provide a better example or discuss this one further. - Training procedure should go before experimental setting. It is simply not its place there, and I was a bit confused when I reached the experimental section without even mentioning training and loss. Please move it earlier into its separate section, it will result in much more natural paper organization. - As much as I liked 90% of the paper, I am a bit disappointed with the results. This section is quite dry and could be improved significantly. - Line 327, "we might expect", is this corroborate with the data set used, or some other data sets? It is definitely intuitive to be expected, but some concrete example and a bit more grounded discussion would help. - Line 336, I do not understand why did the entity count drop? If we only removed unnecessary relations (such as symmetric ones), why would entities be removed as well. This should be elaborated. - Again, the results are very dry, would be good to provide some examples, good and bad cases. This definitely takes away from the quality of presentation. - One suggestion: Line 342, why use "exp" as a suffix, not sure what "exp" refers to? Maybe use "bk" for "background knowledge", or something like that. Not really relevant, just a minor comment. ### ### UPDATE FOLLOWING FEEDBACK ### I would like to thank the authors for their responses. Assuming the authors address my comments (as they said they would) and hopefully add at least a bit of interesting results, this paper definitely belongs to the conference. I have increased my score from 7 to 8 after I read feedback and other reviews.